# The Minimum Number of Strides Required for Reliable Gait Measurements in Older Adult Fallers and Non-Fallers

**DOI:** 10.3390/s24237666

**Published:** 2024-11-30

**Authors:** Drew Commandeur, Marc Klimstra, Kaya Yoshida, Sandra Hundza

**Affiliations:** 1Motion and Mobility Laboratory, University of Victoria, Victoria, BC V8P 5C2, Canada; 2School of Exercise Science, Physical and Health Education, University of Victoria, Victoria, BC V8W 3P2, Canada; 3Canadian Sport Institute Pacific, Victoria, BC V9E 2C5, Canada; 4Rehabilitation Sciences, University of British Columbia, Vancouver, BC V6T 1Z4, Canada; 5Centre for Aging SMART and the GF Strong Rehabilitation Centre, Vancouver, BC V5Z 2G9, Canada

**Keywords:** gait, fallers, dual-task, GAITRite, gait reliability, older adults, minimum strides, difference score

## Abstract

While the value of walking gait metrics collected using pressure-sensing walkways has shown promise for fall risk assessment, there is no consensus on the minimum number of strides required to obtain reliable metrics. This study aimed to determine the minimum stride count required for reliable single-task (ST), dual-task (DT), and difference score (DS) measurements of the spatio-temporal parameters of gait in older adult fallers and non-fallers. Forty community-dwelling older adults (74.6 ± 3.5 years) performed 10 ST and 10 DT walking passes (~100 strides total) across a GAITRite™ pressure mat. Nine truncated datasets (1–9 passes) were created from the original for each walking condition to assess agreement using two-way random effects, absolute agreement, and single-rater intraclass correlations (ICCs). ICCs demonstrated that a minimum of one pass (~10 strides) is sufficient for reliable mean gait metrics for ST and DT conditions and 10–30 strides for DS, while 10–80 strides are needed for reliable gait variability measures, depending on the metric. This study provides stride count recommendations to ensure reliable gait measurement in older adult populations, highlighting that as few as 10–30 strides are necessary for mean metrics, while variability metrics may require up to 80 strides to ensure reliability.

## 1. Introduction

Approximately one in three older adults over the age of 65 falls at least once each year [1], and the severity of falls increases with age [2,3,4], with debilitating consequences including injury, hospitalization, loss of independence, and death [3,4]. Therefore, early identification of those at risk of falls and providing timely, effective interventions is critical [5,6,7]. Fortunately, walking gait metrics have shown promise for predicting future fall risk, and the use of devices such as pressure-sensing walkways can allow for accurate and reliable quantification of gait [8,9,10]. These walkways are portable piezoelectric-based systems of various lengths that can be set up in research and clinical settings and used to collect data from clinical and non-clinical populations. Systems like the GAITRite^TM^ (Version 4.5.7, CIR Systems INC, Sparta, NJ, USA) mat are made of multiple piezoelectric elements that have a spatial resolution of 1–1.5 cm and a temporal resolution of up to 240 Hz, making this device excellent for walking gait spatiotemporal analysis [11]. Research using pressure-sensing walkways, as well as other technologies, has identified differences in several clinically useful gait measurements between fallers and non-fallers in an older adult population [4,5,12,13]. Older adults who fall exhibit slower gait speed, increased stride length variability, decreased swing time, increased stance time, and increased swing time variability [4,14]. Walking gait measures are normally collected during single-task (ST) walking as well as under dual-task (DT) conditions, where a cognitive task like counting backwards is added while walking. It has been shown that DT paradigms are more sensitive than ST walking at distinguishing fallers from non-fallers. Specifically, individuals who fall, especially those with mild cognitive impairment, showed decreased DT gait velocity and a positive association between gait variability and DT complexity [15,16]. Using both ST and DT conditions provides the benefit of being able to assess the intra-individual differences between ST and DT, quantifying the impact of the added cognitive load when multi-tasking. This cognitive load is often calculated as the difference score (DS), which is simply the difference between ST and DT conditions, or it can be calculated as the cognitive cost on gait (CC). The cognitive cost of gait of the DT paradigm can be quantified as the difference between the DT and ST gait metrics divided by the ST and then multiplied by 100. With either the DS or the CC, greater values have been associated with an increased risk of falling [14,17].

While the value of collecting walking gait metrics for ST and DT and the calculation of DS have gained research support as viable clinical fall measurements, there is no consensus on the minimum number of strides required to obtain valid and reliable data for these paradigms with recommendations ranging from 4 to 370 strides depending on the research population and measurement technology [18,19,20,21]. Attempts have been made to quantify the minimum number of walking strides required for reliable gait metrics in healthy young adults, resulting in recommendations of 200 strides to measure mean and standard deviation of ST gait length, width, and timing variables during treadmill walking [19], or 10 strides for mean ST gait metrics and 50 strides for ST gait variability metrics during overland walking [22]. Estimates have also been provided for healthy older adults with mean ST stride velocity and cadence requiring 4 to 9 strides, respectively, DT stride velocity and cadence requiring 9 and 20 strides, and ST velocity variability needing 60 strides. At the same time, it was determined that as many as 370 strides were required for DT velocity variability, as estimated using the Spearman–Brown prophecy formula [18]. In populations with movement disorders (ataxia, essential tremor, and Parkinson’s), only 15–20 strides were needed to achieve excellent consistency for the coefficient of variation of stride time and stride length, while this same study observed that this was consistent with their control group of healthy older adults [21]. While each of these studies helps to provide guidelines of the number of strides that may be required to reliably measure gait variables, the number of gait variables that have been measured is limited. To date, no investigation has determined the minimum number of strides required to reliably determine ST and DT gait metrics, as well as the associated DS in older adult fallers. As DS is calculated as the difference between ST and DT conditions, the errors associated with both ST and DT gait measurement may accumulate in the DS calculation. Therefore, this measurement may have unique requirements when considering accuracy and reliability. Additionally, the specific technologies employed by researchers, including the temporal and spatial resolution of the systems, to measure gait will impact the accuracy and reliability of metrics and potentially the number of strides required. As the GAITRite^TM^ pressure mat is a commonly used device for gait measurement [23], it is, therefore, essential to determine the required number of strides to reliably measure a broad range of gait metrics for ST and DT walking conditions as well as DS using this instrument. Therefore, the aim of this study is to determine the minimum number of strides required to reliably measure spatiotemporal ST, DT gait metrics and DS in healthy older adult fallers and non-fallers using the GAITRite^TM^ pressure mat. We expect that measures of gait variability will require more strides than mean gait metrics for all walking conditions based on previous research [18] as well as known comparative measurement characteristics of variability versus mean [24]. Furthermore, we expect that DT will require more strides than ST given the increased difficulty of the task.

## 2. Materials and Methods

### 2.1. Participants

Forty community-dwelling older adults (74.6 ± 3.5 yrs) were recruited and completed a single- and dual-task walking experiment. Height and weight were measured using a combined stadiometer (cm) and scale (kg) (Health O Meter™). Leg length was measured as the distance from the greater trochanter to the floor using a standard tape measure (cm) while each participant stood upright (without shoes). Leg length was measured to allow for normalization of measures of length and velocity if group differences were observed. In addition, age and self-reported one-year fall history were recorded for each participant. Participants were then classified into two cohorts based on their one-year fall history (fallers N = 25, non-fallers N = 15). Participants recruited for this study were community-dwelling adults aged 65 years or older who spoke fluent English and were able to walk at least 100 m unaided. Participant exclusion criteria included physician-diagnosed dementia or a Mini-Mental Status Examination score (MMSE) below 24, a recent major illness, or a neurological, sensory, or mobility impairment that would impede participation. Participants were screened with the Physical Activity Readiness Questionnaire (PAR-Q), and physician approval was obtained if participants reported ‘yes’ to any question. Participants provided written informed consent, and all procedures were approved by the Human Research Ethics Board at the University of Victoria.

### 2.2. Protocol and Data Preparation

Participants performed 10 single-task (ST) and 10 dual-task (DT) walking passes (approximately 10 strides per pass) wearing regular comfortable walking shoes across a 6.4 m instrumented GAITRite™ pressure-sensing walkway sampling at 120 Hz with an additional 1.5 m acceleration and deceleration zone before and after the walkway. Each pass consisted of one time across the mat and back to the starting point. DT trials consisted of counting backward by serial 7s from a randomly generated 3-digit number and participants were randomized to begin with the DT or ST condition. Measures of gait included step length (cm), stride length (cm), stride width (cm), base of support (cm), stride velocity (cm/s), gait velocity (cm/s), cadence (steps/min), stride time (s), swing% of cycle (%) and swing time (s), as calculated by the GAITRite^TM^ software [25]. The difference in values between the ST and DT conditions was calculated and used to create the difference score (DS) dataset [14,26]. Alongside the complete dataset, comprising 10 passes (approximately 100 strides) for the ST, DT, and DS data, we also created nine truncated datasets for each category. These truncated datasets were derived by removing the last pass, one pass at a time, from the original 10 passes. As a result, each truncated dataset contains between 1 and 9 passes (approximately 10 to 90 strides). The mean and variability (standard deviation) of each gait measure were then calculated for each participant for each dataset. These truncated datasets for ST, DT, and DS were compared to their respective full dataset to assess the level of agreement for mean and variability gait metrics within each of three datasets.

### 2.3. Data Analysis

To assess measurement agreement, variables were grouped into four categories based on the measurement type: length, width, velocity, and timing [14]. For categories, length included mean and variability for step length and stride length, width included mean and variability for base of support and stride width, velocity included mean and variability for stride velocity and velocity and timing included mean and variability for cadence, swing time, stride time and swing % of cycle. Mean and standard deviation (SD) were calculated for each variable for both ST and DT conditions for fallers and non-fallers and then DS was calculated as the difference between ST and DT conditions for the mean and SD. Intraclass correlations were used to assess the absolute agreement for each gait measure across all datasets separately for each cohort (fallers and non-fallers), condition (ST, DT and DS), and measurement type (length, width, velocity, and timing; mean and SD), with reliability coefficients of 0.90 and above considered excellent, and 0.75 and above considered good, with 0.80 commonly used as a standard cut off for reliability, which we have chosen for our analysis [27]. For measures derived from overall trial averages (velocity, cadence, and swing % of cycle) there is no measure of variability for 10 strides as there is a single value reported by GAITRite™ for each pass. All data analysis was performed using custom Python software (Python 3.11, Beaverton, OR, USA), and the Pingouin (version 0.5.4) library was used for two-way random effects, absolute agreement, and single-rater intraclass correlations [28]. Cohort descriptive characteristics are represented as the mean and standard deviation, and cohort differences were determined using Student’s *t*-test (α = 0.05).

## 3. Results

Participant descriptive characteristics, including MMSE, age, height, weight, and leg length, were not significantly different (*p* > 0.05) between fallers and non-fallers. The following values for descriptive statistics will be presented: non-faller mean [standard deviation], faller mean [standard deviation], and *p*-value. Values were MMSE (28.5 [1.2], 28.5 [1.3], 0.36), Age (yrs) (75.7 [3.3], 75.9 [3.3], 0.84), Height (cm) (169.1 [10.5], 166.6 [8.8], 0.43), Weight (kg) (79.1 [15.1], 78.9 [18.5], 0.61), and Leg Length (cm) (91.7 [5.5], 90.5 [5], 0.46). As no differences were observed between the groups in height, weight, or leg length, normalization of length and velocity gait measures was not performed. Mean and SD values for each metric for each pass are presented in Appendix A, Table A1.

### 3.1. Intraclass Correlations of Mean Gait Measures

Comparisons of each mean gait variable separately revealed that both ST and DT walking conditions had an ICC value >0.90 after only one pass of walking for fallers (minimum ICC for ST = 0.947, DT = 0.932) and non-fallers (minimum ICC for ST = 0.959, DT = 0.926). DS mean gait metrics were similarly reliable, with only three metrics requiring more than one pass to achieve a reliability coefficient of 0.80. For the DS base of support, two passes were required to reach the minimum ICC of 0.80 for fallers (ICC = 0.859) and non-fallers (ICC = 0.863). Two walking passes were also required for stride width for fallers (ICC = 0.828) and non-fallers (ICC = 0.844), while for the swing % of a cycle, three passes were required for fallers (0.834) and two passes were required for non-fallers (ICC = 0.919). Table 1 outlines the ICC values by cohort (faller and non-faller) for each pass number within that condition (i.e., Mean and SD for each of ST, DT and DS) for the gait metric that required the highest number of passes to achieve a minimum reliability of ICC > 0.80.

### 3.2. Intraclass Correlations of Variability Gait Measures

For non-fallers in the ST condition, length, width, and cadence variability measures demonstrated agreement by three passes or fewer, while timing and velocity variability measures required more passes to reach ICC > 0.80, as seen in Figure 1.

In the DT walking condition, non-fallers showed measurement agreement by three passes or earlier for all measures. In the ST condition, fallers showed agreement by three passes or earlier for all measures except swing % of cycle variability (five passes), swing time variability (six passes), and velocity variability (five passes). In the DT condition, agreement was achieved by three passes for all measures except the swing % of cycle variability and velocity variability, which each required eight passes (see Figure 1A–D for all variability comparisons). Overall, there was a trend of increasing agreement, with an increasing number of passes with many gait measures only requiring one pass, while DT velocity variability for fallers required eight passes to achieve the target agreement (see Figure 2).

To reach an agreement of ICC >0.80, fallers required three passes for step length, stride length, swing time, stride time, and swing % of cycle variability. Four passes were required for stride velocity and cadence variability. Six passes for stride width and velocity variability, and seven passes for base of support variability. Non-fallers required three passes for swing time and stride time variability. Four passes are required for step length, stride length, and swing % of cycle variability. Five passes were required for the base of support variability and stride width, and eight passes were required for both velocity and cadence variability (see Figure 3).

## 4. Discussion

This is the first study to assess the minimum number of strides required to reliably measure a comprehensive set of single-task (ST) and dual-task (DT) gait metrics, as well as Difference Score (DS) of the dual-task condition on the gait metrics for older adult fallers and non-fallers using the GAITRite™ pressure-sensing mat. This work extends previous research, which examined gait metrics in only non-Faller cohorts using fewer strides or using extrapolated data [18,22]. Current results, based on comparisons to 10 passes of data (~100 strides) demonstrated that for all mean gait metrics, one pass (~10 strides) was required to result in highly reliable measurements for both ST and DT walking for fallers and non-fallers, while DS mean gait metrics required between 1 and 3 passes (~10–30 strides) depending on the gait metric. This presents a highly attainable number of strides to collect in both research and clinical settings.

These findings highlight that the number of strides required to achieve reliability of the measure depends on the specific gait metric, whether the measure is mean or SD, the type of task (i.e., ST or DT) and the cohort characteristics (faller vs. non-faller) Therefore, an investigator must consider these factors when determining the appropriate number of strides for their protocol. For example, if timing variability or velocity variability measures are used, particularly for DS metrics, then as many as 8 passes (~80 strides) may be required for reliable measures. The greatest number of strides were required for DS variability metrics, with non-fallers requiring 3–8 passes (~30–80 strides) and fallers needing 3–7 passes (~30–70 strides) to ensure reliability. ST gait variability measures required 1–6 passes (~10–60 strides) for fallers and non-fallers, DT variability measures required 1–3 passes (~10–30 strides) for fallers and 1–8 passes (~10–80 strides) for non-fallers. However, if the required standard for reliability was reduced to an ICC value of 0.75, which is still considered good reliability, fewer strides would be required for most variability metrics.

As clearly demonstrated by Estabook et al. [24], mean measures are more reliable than variability measures in general. Therefore, it was not surprising that to achieve acceptable reliability, the mean gait metrics required fewer strides than most measures of gait variability. Overall, ST and DT mean gait measures required a similar number of strides to achieve acceptable reliability. However, it was unexpected that ST SD measures required more strides than DT SD measures to achieve reliability (Table 1). It is speculated that as participants performed the DT, they were highly focused and performing at the limit of their capacity, which could constrain variability as they are operating near the limits of their ability. This may have produced less variability, whereas during ST, more variability of performance was observed from stride to stride based on the wider range of available gait strategies. For mean gait measures, as expected, overall, fallers required more strides than non-fallers to achieve reliability, while for SD gait measures, non-fallers required more strides than fallers to achieve reliability. There is a small sample in each cohort, and it is speculated that there were potentially unique characteristics within each of these cohorts that could contribute to this unexpected result.

### 4.1. Stride Count Recommendations

Excellent reliability was observed for all ST and DT mean gait measures, requiring only 10 strides and DS gait measures 10–30 strides, while measures of gait variability required more strides to achieve the desired level of reliability (10 to 80 strides). Consistent with the findings of others, there is a clear trend showing that increased reliability for measures of gait variability is achieved with an increased stride count [10,18]. Hollman et al., 2010 [18] provided stride count recommendations for characterising walking gait based on assessments of the reliability of DT stride velocity (ICC = 0.93), cadence (ICC = 0.83) and stride velocity variability (ICC = 0.23) in healthy older adults. Based on these results, they determined that to obtain reliable measures of mean stride velocity, nine strides were required, 20 strides were required for mean cadence, and as many as 370 strides were required for stride velocity variability. While we observed similar results for mean stride velocity and cadence in the ST and DT conditions, substantially fewer strides were required for stride velocity variability for all conditions in both cohorts (10–80 strides) than what was observed by Hollman et al., 2010 (370 strides) [18]. This discrepancy could be due to the differences in stride velocity variability calculation between studies. Hollman et al. (2010) calculated stride velocity variability as the average coefficient of variation over the whole trial, while in the present study, stride velocity variability was calculated as the average stride-to-stride standard deviation. Additionally, Hollman et al. only measured 14 strides for each participant (N = 24) in the DT condition and used the Spearman–Brown prophecy formula to extrapolate reliability coefficients for a given stride count. Similar to our findings, König et al., (2014) [22] employed ICC (2,1) analysis and found that only 10 strides were required for mean gait parameters (ICC = 0.88 (10 strides) to ICC = 0.98 (60 strides)) and that increasing stride counts resulted in improved reliability for variability measurements up to 50 strides (ICC = 0.60 (10 strides) to ICC = 0.90 (60 strides). Importantly, König et al., (2014) directly measured 60 strides and the results were not estimated from extrapolated data [22]. In addition to measuring the reliability of gait metric estimation from adequate stride counts, several studies have assessed test–retest reliability of gait metrics and found that mean gait metrics were more reliable than measures of variability [18,29]. Almarwani et al. 2016 [30] examined gait measurement test–retest reliability in healthy older adults using ICC (2,1) and found that mean measures had excellent test–retest reliability, while measures of variability were fair to good. Reliability for mean gait measures performs well, while variability provides lower intraclass correlations. This may be particularly due to gait due to its fractal nature, which leads to non-stochastic variability, whereby the variability of each successive stride is related to the previous stride history, requiring a sufficient number of strides to achieve consistency [24,31]. While no previous attempt has been made to determine the minimum number of strides required to measure gait in fallers, particularly for DT or DS metrics, there is evidence supporting increased variability in fallers, which could require more strides to achieve consistent measurements [8,32].

The number of strides chosen by a clinician or researcher would be dependent on the specific objectives of their investigation. For example, if the intended use for the data was not fully defined or if many gait metrics were required, then it would be prudent to select the number of passes associated with the gait metric, descriptive statistic, cohort and the condition that required the highest number of passes to achieve a minimum reliability of ICC > 0.80 as outlined in Table 1. Doing this ensures the reliability of all metrics for all scenarios that may be required by the researcher or clinician. Further, Table 1 indicates the minimum number of passes to achieve ICC > 0.80 as well as the number of passes at which gathering further data does not improve the reliability of the data, which is indicated by the stabilization of the ICC value with successive trials. Lastly, if the cohort, condition, descriptive statistic and specific gait metrics of interest are known, then Table 2 (see below) outlines the required the number of passes to achieve a minimum reliability of ICC > 0.80. Overall, when using the GAITRite^TM^ system, eight passes (~80 strides) will achieve sufficient reliability for older adult fallers and non-fallers for all gait metrics for measures of mean and variability in all conditions.

When determining the reliability of gait metrics for ST, DT and the DS calculation, it is important to consider the statistical relationship between these measures, especially when employing the intraclass correlation coefficient (ICC) as a reliability statistic. ST and DT are independent conditions, and their respective ICC values provide insights into the consistency of measurements within each condition. However, DS is a metric that is derived from the difference between ST and DT measures. Therefore, there are two possible considerations for determining the reliability of DS metrics. Firstly, the reliability of DS metrics can be inferred if ST and DT measures are reliable, as the resulting derived DS measures could be considered inherently reliable, regardless of whether the intraclass coefficients comparing full and reduced datasets are high. Secondly, if using the reliability of DS metrics to determine the minimum number of strides, it is important to consider that since DS measures are dependent on the reliability of the primary metrics (ST and DT) from which it is derived, errors or variability present in either ST or DT metrics may propagate into the DS metrics, potentially compounding errors and reducing the reliability of the DS metrics. Additionally, in the cases where ST and DT metrics are highly correlated and potentially produce small differences across participants, the variance of the DS metrics may be limited, which may result in a lower ICC for these derived metrics. Conversely, high variability in the DS metrics may increase the ICC and could reflect noise rather than true variability in task performance. Therefore, when presenting ICCs values for both the primary (ST and DT) and derived (DS) metrics, it is important to interpret the results in the context of these interdependencies. This is why the ICC for DS metrics was considered alongside the ICCs of the ST and DT metrics.

### 4.2. Limitations

Limitations of our study include the limited spatial resolution of the GAITRite™ pressure mat, which may increase the variability of measurements. Our recommendation for a minimum number of strides is limited to overland walking on a smooth, flat surface, and footwear may have an influence on the number of strides required to reach a gait state, which is a consideration when designing research protocols [33]. Gait data in the current study were collected during non-continuous walking due to constraints of GAITRite™ mat length, therefore, findings may be different with a continuous walking path or with the utilization of alternate measurement technologies.

## 5. Conclusions

We found that a minimum of three overland walking passes (~30 strides) are required for reliable measurement of most gait measures in older adult fallers and non-fallers. If only measures of mean ST or DT gait metrics are being measured, then as few as one pass (~10 strides) is required. If certain gait metrics such as DS velocity variability or cadence variability are desired, then as many as eight passes (~80 strides) may be warranted. Future research should investigate the minimum number of strides to reliably measure gait in older adult fallers and non-fallers using alternate data collection techniques, including inertial measurement units, 3D motion capture, 2-D video motion capture, and treadmill walking. Continuous overland walking should also be examined; however, technical limitations of systems such as GAITRite™ or optoelectronic 3D motion capture limit them from this analysis.

## Figures and Tables

**Figure 1 sensors-24-07666-f001:**
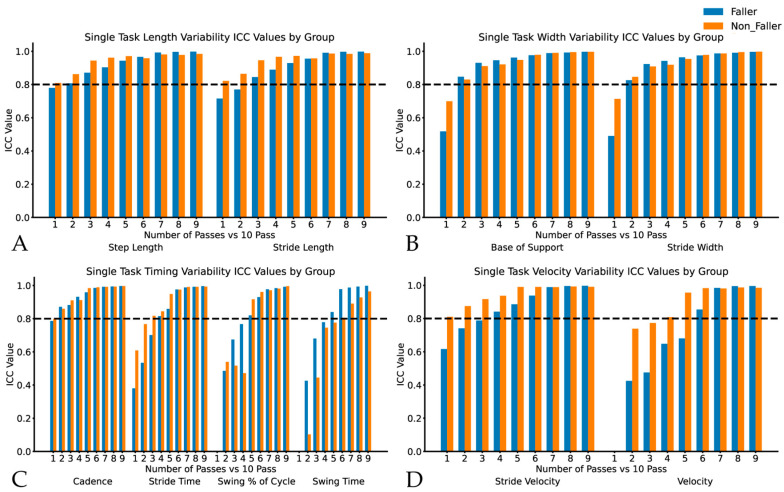
Intraclass correlations for single-task gait variability metrics in fallers and non-fallers displaying variability for Panel (**A**): step and stride length; Panel (**B**): base of support and stride width; Panel (**C**): cadence, stride time, swing % of cycle and swing time; and Panel (**D**): stride velocity and velocity. ICC values represent the comparison between each truncated dataset (1–9 pass) vs. the complete 10-pass dataset.

**Figure 2 sensors-24-07666-f002:**
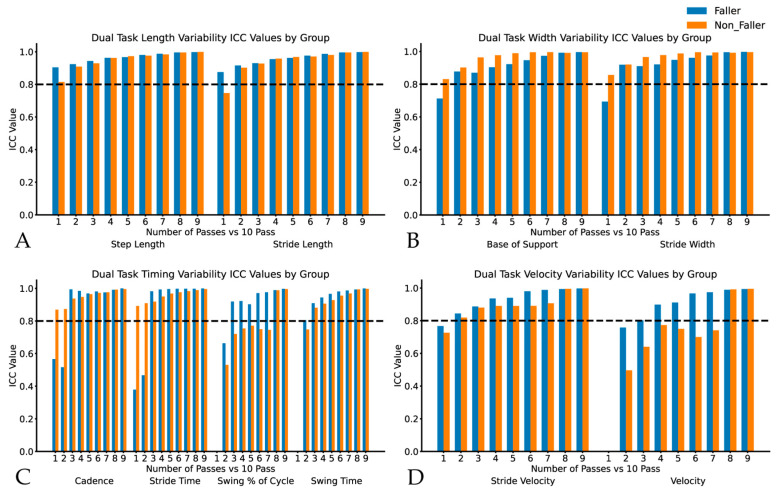
Intraclass correlations for dual-task gait variability metrics in fallers and non-fallers displaying variability for Panel (**A**): step and stride length, Panel (**B**): base of support and stride width, Panel (**C**): cadence, stride time, swing % of cycle and swing time, and Panel (**D**): stride velocity and velocity. ICC values represent the comparison between each truncated dataset (1–9 pass) vs. the complete ten-pass dataset.

**Figure 3 sensors-24-07666-f003:**
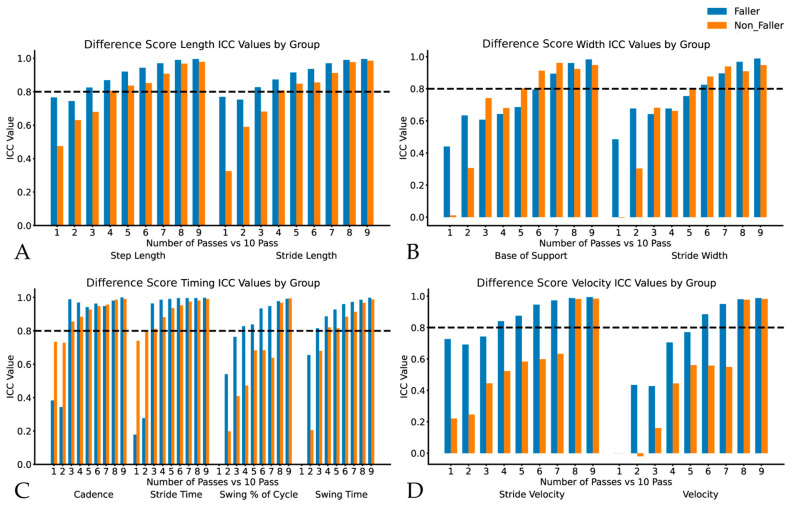
Intraclass correlations for difference score gait variability metrics in fallers and non-fallers displaying variability for Panel (**A**): step and stride length; Panel (**B**): base of support and stride width; Panel (**C**): cadence, stride time, swing % of cycle and sing time; and Panel (**D**): stride velocity and velocity.

**Table 1 sensors-24-07666-t001:** Gait metric requiring the greatest number of passes to reach ICC = 0.8 compared to 10 passes for older adult fallers and non-fallers walking across a GAITRite^TM^ pressure mat during dual-task and single-task conditions, as well as the Difference. Missing values are present for metrics with no variability for 1 pass of walking gait due to GAITRite^TM^ calculation for that metric. ICC values > 0.80 are bolded.

Condition	Group	Gait Metric	Number of Passes								
			1	2	3	4	5	6	7	8	9
Dual Task (Mean)	Faller	Stride Time	**0.92**	0.91	0.96	0.97	0.97	0.99	0.99	1.00	1.00
	Non-Faller	Swing % of Cycle	**0.95**	0.98	0.99	1.00	1.00	1.00	1.00	1.00	1.00
Single Task (Mean)	Faller	Base of Support	**0.91**	0.97	0.98	0.99	1.00	1.00	1.00	1.00	1.00
	Non-Faller	Base of Support	**0.96**	0.99	0.99	0.99	1.00	1.00	1.00	1.00	1.00
Difference Score (Mean)	Faller	Swing % of Cycle	0.7	0.72	**0.83**	0.9	0.92	0.96	0.97	0.98	1.00
	Non-Faller	Stride Width	0.6	**0.84**	0.9	0.94	0.94	0.97	0.99	0.99	1.00
Dual Task (SD)	Faller	Swing Time	0.38	0.47	**0.98**	0.99	1.00	1.00	1.00	1.00	1.00
	Non-Faller	Velocity		0.5	0.64	0.77	0.75	0.7	0.74	**0.99**	1.00
Single Task (SD)	Faller	Velocity		0.43	0.48	0.65	0.68	**0.85**	0.98	1.00	1.00
	Non-Faller	Swing % of Cycle		0.1	0.45	0.75	0.78	**0.81**	0.89	0.93	0.96
Difference Score (SD)	Faller	Base of Support	0.44	0.63	0.61	0.64	0.69	0.79	**0.89**	0.96	0.98
	Non-Faller	Velocity		0.00	0.16	0.44	0.56	0.56	0.55	**0.98**	0.98

**Table 2 sensors-24-07666-t002:** Recommended minimum stride to ensure reliable mean variability gait measures during single-task, dual-task, and difference score walking across a GAITRite^TM^ pressure mat for older adult fallers and non-fallers. Values in bold denote variables that required more than the baseline of 10 strides.

		Non-Fallers (Minimum Stride Count)			Fallers (Minimum Stride Count)		
	Gait Variable	Single-Task	Dual-Task	Difference Score	Single-Task	Dual-Task	Difference Score
Mean	Step Length	10	10	10	10	10	10
	Stride Length	10	10	10	10	10	10
	Base of Support	10	10	**20**	10	10	**20**
	Stride Width	10	10	**20**	10	10	**20**
	Cadence	10	10	10	10	10	10
	Stride Time	10	10	10	10	10	10
	Swing Time	10	10	10	10	10	10
	Swing % of Cycle	10	10	**20**	10	10	**30**
	Stride Velocity	10	10	10	10	10	10
	Velocity	10	10	10	10	10	10
Variability	Step Length	10	10	**40**	20	10	**30**
	Stride Length	10	**20**	**40**	**30**	**20**	**30**
	Base of Support	**20**	10	**50**	**20**	**20**	**70**
	Stride Width	**20**	10	**50**	**20**	**20**	**60**
	Cadence	**50**	**80**	**80**	**50**	**30**	**40**
	Stride Time	**20**	10	**30**	**20**	**30**	**30**
	Swing Time	**30**	10	**30**	**40**	**30**	**30**
	Swing % of Cycle	**60**	**30**	**40**	**50**	**20**	**30**
	Stride Velocity	10	**20**	**80**	**40**	**20**	**40**
	Velocity	**40**	**80**	**80**	**60**	**30**	**60**

## Data Availability

Data used in this study are not able to be shared outside of the original research group.

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
