# Peer review of "The Minimum Number of Strides Required for Reliable Gait Measurements in Older Adult Fallers and Non-Fallers"

_sensors, 2024, doi:10.3390/s24237666_

Round 1

Reviewer 1 Report

Comments and Suggestions for Authors

The present manuscript aims to determine the minimum number of strides required to reliably measure spatiotemporal gait metrics in healthy older adults with and without falls. It also takes into consideration whether walking is performed in a simple or single-task fashion, but also in a compound-task fashion and the differences between the tasks. The main material used was the GAITRiteTM plantar pressure platform.

Generic comments:

This study provides valuable information that could be used in both research and clinical settings in order to define future protocols for reliable analysis. Based on the minimum number of reliable strides for spatiotemporal gait variables.

Although the design is well thought out, there are a number of crucial issues that need to be addressed regarding the methodology of the study. In particular, the main study variables (these spatio-temporal gait variables) and their treatment are not mentioned and need to be clarified. In addition, some crucial aspects of the statistics used and the treatment of the data need to be clarified.

Please respond point by point to specific comments to clarify this manuscript.

Specific comments:

Abstract

Lines 17-18: What are the variables under study?

Lines 24-25: Where it says: “highlighting that fewer strides are…” Please be more precise, you could mention after how many strides metrics are reliable for mean metrics.

Introduction

Lines 61-64: The range mentioned above is very wide, it would be convenient to keep the same structure and add the stride ranges recommended by these studies depending on the age of the sample studied in each study. This could help to understand why the range is so wide.

Lines 75-78: Is GAITRite a gold standard? No evidence is mentioned in this regard. Please clarify this information in the introduction or rephrase this statement.

Line 80: What is your hypothesis?

Methods

Line 85: Before applying exclusion criteria, you should establish what criteria have been established to select participants. Please add inclusion criteria.

Line 93: Did the participants walk with shoes on or did they walk barefoot? Please add this information.

Line 95: I am missing the full list and definitions of the variables used for this study. It is crucial that the variables are described.

Line 100: “The difference in values…”? What are you measuring? What values? Swing phase (time), stance phase (time)?  Step Length (meters), stride length (meters)?

Line 102: What do you mean by “nine truncated data”? Clarify this please.

Line 105: By “gait measure” do you mean “pass”?

Line 114: What ICC have been used? Please clarify this.

Line 114: Here the authors mention that ICC was used for assessing the absolute agreement for each cohort, condition and measurement type. Is this statement correct? During results I only see tables of ICC related to measurement type per pass. ICC was not used for comparing between cohorts’ participants (faller and non-faller) or between tasks (ST vs DT vs DS). Please clarify this. The only absolute agreement studied seems to be intra spatiotemporal parameters per pass and inter passes. And as mentioned in line 124, a t-test was used to compare between cohorts.

Line 116: This is the first time I see the parameters or variables used during the manuscript. They should be described beforehand.

Line 120: More variables? “Velocity, cadence, percentage of swing phase of cycle”?

Lines 124-125: Where are the results of the t-test in the results section?

Results

Line 127: No material has been described for measuring ‘height, weight or leg length’. This is secondary to the proposed study objective but should be described in the methods. Descriptive variables and materials should also be mentioned beforehand.

Line 134: What is the mean and SD of these parameters per passes? One table with these parameters should be facilitated for the conditions of this study (ST, DT and DS) and per passes. This table can be found as an annexe if you wish.

Line 144, Table 2: what is “cognitive cost”? Do you mean DS (difference score)? Please, be consistent with the terms used throughout the manuscript.

Line 144, Table 2: What does this variable include: the average of the ICCs of all spatio-temporal metrics? What for? It could confuse the reader. This table does not provide much information because it is the average of the ICCs, confusing information. I would leave it for the end of the results or remove it.

Lines 183-186: Please be concise. And avoid value judgements.

Figures 1, 2 and 3: “Number of passes vs 10 pass” has been repeated at the bottom of each graph. What does it mean? Could you rephrase it or explain it? Do you mean “Number of passes vs. 10 passes”?

Discussion

Lines 207-210: What do you mean? Rephrase this sentence.

Line 215: Why ‘as expected’? During the introduction the authors did not state any evidence-based hypotheses. If this is a value judgement, please delete ‘as expected’.

Lines 217-222: After this brief summary of the results, I miss a discussion of why for the variability the expected values were higher, or why the ST condition needs more passes than the DT condition. There is no discussion on the subject.

Line 229: Why do some studies claim that 370 strides are needed to study variability in stride velocity? Why is this value so different from the results of the present study?

Lines 244-245: First information I have seen on how a single variable has been calculated. It is crucial that this information appears in the material and methods section so that all readers understand how this study has been carried out.

Lines 250 and 259: The authors mention several times in the discussion which ICC was used for each study. As I mentioned earlier, the authors should mention which ICCs were used for the present study.

Author Response

Report 1

Comments and Suggestions for Authors

The present manuscript aims to determine the minimum number of strides required to reliably measure spatiotemporal gait metrics in healthy older adults with and without falls. It also takes into consideration whether walking is performed in a simple or single-task fashion, but also in a compound-task fashion and the differences between the tasks. The main material used was the GAITRiteTM plantar pressure platform.

Generic comments:

This study provides valuable information that could be used in both research and clinical settings in order to define future protocols for reliable analysis. Based on the minimum number of reliable strides for spatiotemporal gait variables.

Although the design is well thought out, there are a number of crucial issues that need to be addressed regarding the methodology of the study. In particular, the main study variables (these spatio-temporal gait variables) and their treatment are not mentioned and need to be clarified. In addition, some crucial aspects of the statistics used and the treatment of the data need to be clarified.

Please respond point by point to specific comments to clarify this manuscript.

Specific comments:

Abstract

Lines 17-18: What are the variables under study?

We thank the reviewer for this comment and have added the variables under study in the abstract

Lines 24-25: Where it says: “highlighting that fewer strides are…” Please be more precise, you could mention after how many strides metrics are reliable for mean metrics.

Thank you. We have now included how many strides are needed in this section.

Introduction

Lines 61-64: The range mentioned above is very wide, it would be convenient to keep the same structure and add the stride ranges recommended by these studies depending on the age of the sample studied in each study. This could help to understand why the range is so wide.

Thank you for this comment, we have added more detail in this section to clarify why the range is so large, which is mostly based on the nature of the studies to determine them.

Lines 75-78: Is GAITRite a gold standard? No evidence is mentioned in this regard. Please clarify this information in the introduction or rephrase this statement.

Thank you for this comment, we have provided an appropriate reference for this statement.

Line 80: What is your hypothesis?

We have added our hypothesis near here

Methods

Line 85: Before applying exclusion criteria, you should establish what criteria have been established to select participants. Please add inclusion criteria.

Thank you for this comment, we have added inclusion criteria.

Line 93: Did the participants walk with shoes on or did they walk barefoot? Please add this information.

We have added this information.

Line 95: I am missing the full list and definitions of the variables used for this study. It is crucial that the variables are described.

We have provided a list of variables and reference to the GAITRite technical manual that includes the detailed description of how variables are calculated.

Line 100: “The difference in values…”? What are you measuring? What values? Swing phase (time), stance phase (time)?  Step Length (meters), stride length (meters)?

Similar to the comment above we have added this information including units.

Line 102: What do you mean by “nine truncated data”? Clarify this please.

We have provided additional clarification for this process.

Line 105: By “gait measure” do you mean “pass”?

We have clarified this to refer to each gait variable.

Line 114: What ICC have been used? Please clarify this.

This has been updated to include the ICC type

Line 114: Here the authors mention that ICC was used for assessing the absolute agreement for each cohort, condition and measurement type. Is this statement correct? During results I only see tables of ICC related to measurement type per pass. ICC was not used for comparing between cohorts’ participants (faller and non-faller) or between tasks (ST vs DT vs DS). Please clarify this. The only absolute agreement studied seems to be intra spatiotemporal parameters per pass and inter passes. And as mentioned in line 124, a t-test was used to compare between cohorts.

The text accurately reflects that intraclass correlations were performed separately for each cohort, condition, and measurement type, and there is no indication that comparisons were made or intended between groups. “Intraclass correlations were used to assess the absolute agreement for each gait measure across all data sets separately for each cohort (Fallers and Non-Fallers), condition (ST, DT and DS), and measurement type (length, width, velocity, and timing; mean and SD)”

Line 116: This is the first time I see the parameters or variables used during the manuscript. They should be described beforehand.

Thank you for this point. Variables are now appropriately introduced with a reference provided for their definitions.

Line 120: More variables? “Velocity, cadence, percentage of swing phase of cycle”?

Thank you, we have addressed this concern in the methods section along with the comment above.

Lines 124-125: Where are the results of the t-test in the results section?

An in-text p-value has been added to supplement Table 1.

Results

Line 127: No material has been described for measuring ‘height, weight or leg length’. This is secondary to the proposed study objective but should be described in the methods. Descriptive variables and materials should also be mentioned beforehand.

This is now appropriately described in the introduction, and the result from these measures is provided in lines 165-166.

Line 134: What is the mean and SD of these parameters per passes? One table with these parameters should be facilitated for the conditions of this study (ST, DT and DS) and per passes. This table can be found as an annexe if you wish.

This table has been added as an Appendix A Table A.1 at the end of the paper.

Line 144, Table 2: what is “cognitive cost”? Do you mean DS (difference score)? Please, be consistent with the terms used throughout the manuscript.

We thank the reviewer for catching this oversight, it has been corrected to appropriately reflect that these are difference scores throughout the manuscript. Further, we have decided to remove this DS section in this table to streamline the results and for clarity.

Line 144, Table 2: What does this variable include: the average of the ICCs of all spatio-temporal metrics? What for? It could confuse the reader. This table does not provide much information because it is the average of the ICCs, confusing information. I would leave it for the end of the results or remove it.

We thank the reviewer for this suggestion. We have simplified the table and suggest it’s use to describe the basic trend of increasing reliability with additional strides, and the differences between mean and variability gait variables.

Lines 183-186: Please be concise. And avoid value judgements.

The results have been re-worded for clarity and brevity.

Figures 1, 2 and 3: “Number of passes vs 10 pass” has been repeated at the bottom of each graph. What does it mean? Could you rephrase it or explain it? Do you mean “Number of passes vs. 10 passes”?

Thank you for pointing out the potential for misinterpretation. Clarification has been provided in the figure descriptions.

Discussion

Lines 207-210: What do you mean? Rephrase this sentence.

This sentence has been revised for clarity.

Line 215: Why ‘as expected’? During the introduction the authors did not state any evidence-based hypotheses. If this is a value judgement, please delete ‘as expected’.

We have provided the appropriate context for this statement.

Lines 217-222: After this brief summary of the results, I miss a discussion of why for the variability the expected values were higher, or why the ST condition needs more passes than the DT condition. There is no discussion on the subject.

Thank you for raising this point. We have included our rationale for these findings in the the discussion as outlined here. As clearly demonstrated by Estabook and colleagues, mean measures are more reliable than variability measures. Therefore, it was not surprising that the mean gait metrics required fewer strides thanmost measures of gait variability to produce reliable measures (Table 2). Overall, ST and DT mean gait measures required similar number of strides to achieve acceptable reliability. However, it was unexpected that ST SD measures required more strides than DT SD measures to achieve reliability (Table 2). It is speculated that as participants performed the DT, they were highly focused performing at the limit of their capacity within a narrow range at the top limits of their function. This may have produced less variability, whereas during ST more variability of performance was observed stride to stride based on the wider range available. For mean gait measures, as expected, overall, Fallers required more strides than Non-Fallers to achieve reliability, while for SD gait measures Non-Fallers required more strides than Fallers to achieve reliability. There is a small sample in each cohort and and it is speculated that there were potentially unique characteristics within each of these cohorts that is responsible for this unexpected result.  

 Line 229: Why do some studies claim that 370 strides are needed to study variability in stride velocity? Why is this value so different from the results of the present study?

This justification is provided in the following sentences describing the difference in the calculation of variability, as well as Hollman’s use of a limited dataset (14 strides) which were then extrapolated using the Spearman-Brown prophecy formula, while our study measured more actual strides.

Lines 244-245: First information I have seen on how a single variable has been calculated. It is crucial that this information appears in the material and methods section so that all readers understand how this study has been carried out.

Thank you, we have appropriately introduced the variables and provided a reference for their calculations.

Lines 250 and 259: The authors mention several times in the discussion which ICC was used for each study. As I mentioned earlier, the authors should mention which ICCs were used for the present study.

Thank you for identifying this oversight, the ICC used is now reported in both the methods and results section.

Reviewer 2 Report

Comments and Suggestions for Authors

The manuscript provided suggestions on the minimum strides numbers that can be used to achieve reliable measurements of different gait metrics and the corresponding difference scores in healthy older adults fallers (1-year record) and non-fallers group. The manuscript provided enough sample sizes and reasonable analysis. The reviewer assumes this work can be accepted after addressing the following concerns.

11.       The author used one-year fall history to distinguish the Fallers and Non-Fallers groups. Did the authors consider potential causes of the fall(s) in the Faller group? Will the participants caused by non-personal reasons (eg. environment conditions) sufficient enough to be considered as the Faller group?

22.       To achieve an acceptable measurement agreement, why do the Non-Fallers need more passes compared with Fallers in the DS variability measurement (line 139) and in some gait variability metrics during dual-task?

33.       For each participant, the mean and standard deviation of each gait measure were calculated based on nine truncated data sets that derived from one independent test. To obtain a more reliable datasets, is it possible to conduct three independent experiments at separate times for each participant?

Minor:

1.       The authors are encouraged to provide sufficient descriptions in the content. For example, in line 162, “while timing and velocity variability measures required more passes”.

Author Response

Report 2

Comments and Suggestions for Authors

The manuscript provided suggestions on the minimum strides numbers that can be used to achieve reliable measurements of different gait metrics and the corresponding difference scores in healthy older adults fallers (1-year record) and non-fallers group. The manuscript provided enough sample sizes and reasonable analysis. The reviewer assumes this work can be accepted after addressing the following concerns.

  1. The author used one-year fall history to distinguish the Fallers and Non-Fallers groups. Did the authors consider potential causes of the fall(s) in the Faller group? Will the participants caused by non-personal reasons (eg. environment conditions) sufficient enough to be considered as the Faller group?

The circumstances around each fall were carefully reviewed with the participants to determine that they were not a result of environmental conditions (e.g. an icy surface with unexpected perturbations) or as a result of an acute medical conditions (e.g. fainting). Only if factors such as these were not at play was it considered a fall.

  1. To achieve an acceptable measurement agreement, why do the Non-Fallers need more passes compared with Fallers in the DS variability measurement (line 139) and in some gait variability metrics during dual-task?

For mean gait measures, as expected, overall, Fallers required more strides than Non-Fallers to achieve reliability, while for SD gait measures Non-Fallers required more strides than Fallers to achieve reliability. There is a small sample in each cohort and and it is speculated that there were potentially unique characteristics within each of these cohorts that is responsible for this unexpected result.   This has been added to the paper.

  1. For each participant, the mean and standard deviation of each gait measure were calculated based on nine truncated data sets that derived from one independent test. To obtain a more reliable datasets, is it possible to conduct three independent experiments at separate times for each participant?

We thank the reviewer for their suggestion. Unfortunately, due to the difficulty of conducting research with this population, it is not possible to repeat this assessment multiple times for each participant. We feel that our dataset provides a robust sample that is already richer than what has been reported previously in other research.

Minor:

  1. The authors are encouraged to provide sufficient descriptions in the content. For example, in line 162, “while timing and velocity variability measures required more passes”.

Thank you for this feedback. We have gone through the manuscript and attempted to add more description in sentences like the one mentioned.

Round 2

Reviewer 1 Report

Comments and Suggestions for Authors

General comments,

Thank you for your response and for the changes made. The manuscript has been significantly improved.

However, there are some minor specific comments that may be reviewed before this manuscript is accepted. And a major concern about Table 2 and the information related to this table, where an average of ICCs is calculated from all gait measures. As the variables used have different units of measurement and magnitudes, it is not correct to average the ICC values obtained from the gait variables. This information must be removed from the results.

Specific comments

Introduction

Line 88-89: The reference provided is not a validation of GAITRite® with a gold standard (i.e. a 3D motion capture system or similar) to be validated. The reference is a comparison of two GAITRite® models. Therefore, it can't be said that GAITRite® is a gold standard, please rephrase this sentence.

Line 98: Add a full stop for the subsection.

Material and methods

Lines 101-102: What were the instruments or materials used to collect “height, weight, and leg length”? How did you measure them?

Line 131: Use “gait measure” instead of “gait variable”, due to, in line 121 you mention “Measures of gait”.

Lines 136-137: What parameters were selected for the category “timing”? Further information must be provided about what variables is allocated into each category.

Results

Table 2 and lines 161-170: Remove Table 2 and any information where you use the average ICC values with all parameters at once. It is not correct to average ICC values obtained from different variables, especially if these variables have different units of measurement or represent different aspects. The ICC is not a linear measure and averaging the ICC values of different variables may give a misleading result. Please remove this section. This manuscript has many results and findings with a specific analysis per variable from line 174 onwards.

Discussion

Lines 258-271: Remove the italic style from the entire paragraph.

Line 271: Add a full stop for the subsection.

Author Response

General comments,

Thank you for your response and for the changes made. The manuscript has been significantly improved.

However, there are some minor specific comments that may be reviewed before this manuscript is accepted. And a major concern about Table 2 and the information related to this table, where an average of ICCs is calculated from all gait measures. As the variables used have different units of measurement and magnitudes, it is not correct to average the ICC values obtained from the gait variables. This information must be removed from the results.

We thank the reviewer for this comment and we have removed the current table 2 and any references to this table in the document.

Specific comments

Introduction

Line 88-89: The reference provided is not a validation of GAITRite® with a gold standard (i.e. a 3D motion capture system or similar) to be validated. The reference is a comparison of two GAITRite® models. Therefore, it can't be said that GAITRite® is a gold standard, please rephrase this sentence.

We thank the reviewer for this comment and we have modified the phrasing of the sentence.

Line 98: Add a full stop for the subsection.

We have added a full stop for this subsection.

Material and methods

Lines 101-102: What were the instruments or materials used to collect “height, weight, and leg length”? How did you measure them?

We have added the instruments used to collect height, weight and leg length.

Line 131: Use “gait measure” instead of “gait variable”, due to, in line 121 you mention “Measures of gait”.

We have modified this sentence to use gait measure instead of gait variable.

Lines 136-137: What parameters were selected for the category “timing”? Further information must be provided about what variables is allocated into each category.

We have added more details and specifics around the category timing. 

Results

Table 2 and lines 161-170: Remove Table 2 and any information where you use the average ICC values with all parameters at once. It is not correct to average ICC values obtained from different variables, especially if these variables have different units of measurement or represent different aspects. The ICC is not a linear measure and averaging the ICC values of different variables may give a misleading result. Please remove this section. This manuscript has many results and findings with a specific analysis per variable from line 174 onwards.

Again, we thank the reviewer for this comment and we have removed the current table 2 and any references to this table in the document.

Discussion

Lines 258-271: Remove the italic style from the entire paragraph.

We have removed the italic style for the paragraph.

Line 271: Add a full stop for the subsection.

We have added a full stop for the paragraph. We also noticed an issue with this formating which may be dealt with with support from the editor.